# Plant Promoters: Their Identification, Characterization, and Role in Gene Regulation

**DOI:** 10.3390/genes14061226

**Published:** 2023-06-06

**Authors:** Liliana Villao-Uzho, Tatiana Chávez-Navarrete, Ricardo Pacheco-Coello, Eduardo Sánchez-Timm, Efrén Santos-Ordóñez

**Affiliations:** 1Biotechnological Research Center of Ecuador, ESPOL Polytechnic University, Escuela Superior Politécnica del Litoral, ESPOL, Gustavo Galindo Campus Km. 30.5 Vía Perimetral, Guayaquil 090902, Ecuador; lilelvil@espol.edu.ec (L.V.-U.); tachavez@espol.edu.ec (T.C.-N.); rhpachec@espol.edu.ec (R.P.-C.); lesanche@espol.edu.ec (E.S.-T.); 2Faculty of Life Sciences, ESPOL Polytechnic University, Escuela Superior Politécnica del Litoral, ESPOL, Gustavo Galindo Campus Km. 30.5 Vía Perimetral, Guayaquil 090902, Ecuador

**Keywords:** transcription, gene expression, genetic engineering, genetics, promoters

## Abstract

One of the strategies to overcome diseases or abiotic stress in crops is the use of improved varieties. Genetic improvement could be accomplished through different methods, including conventional breeding, induced mutation, genetic transformation, or gene editing. The gene function and regulated expression through promoters are necessary for transgenic crops to improve specific traits. The variety of promoter sequences has increased in the generation of genetically modified crops because they could lead to the expression of the gene responsible for the improved trait in a specific manner. Therefore, the characterization of the promoter activity is necessary for the generation of biotechnological crops. That is why several analyses have focused on identifying and isolating promoters using techniques such as reverse transcriptase-polymerase chain reaction (RT-PCR), genetic libraries, cloning, and sequencing. Promoter analysis involves the plant genetic transformation method, a potent tool for determining the promoter activity and function of genes in plants, contributing to understanding gene regulation and plant development. Furthermore, the study of promoters that play a fundamental role in gene regulation is highly relevant. The study of regulation and development in transgenic organisms has made it possible to understand the benefits of directing gene expression in a temporal, spatial, and even controlled manner, confirming the great diversity of promoters discovered and developed. Therefore, promoters are a crucial tool in biotechnological processes to ensure the correct expression of a gene. This review highlights various types of promoters and their functionality in the generation of genetically modified crops.

## 1. Introduction

### 1.1. Promoters and Their Importance in Genetically Modified Crops

Sustainable maintenance in agriculture with a high productivity level has also been thanks to advances in plant biotechnology, especially through the implementation of genetically modified crops. Therefore, the search and availability of new promoters have been a high priority among researchers since they provide spatial and temporal control in transgene expression. Over the years, various promoters have been identified in different organisms for use in genetically modified plants because they are a potent tool in regulating gene expression leading to improved traits [1]. The success of gene expression depends to a great extent on the use and efficient selection of promoters, even being able to drive multiple transgenes by the same promoter in a single plant [2,3].

Promoters are located in the 5′ region of a gene and are composed of a specific nucleotide sequence controlling the expression of DNA in a physical, adjacent, and functional manner. In this way, gene regulation mainly depends on these essential regulatory elements for transcription initiation [1]. A DNA sequence located at the 5′ end of the coding region of a gene, which includes different binding regions for transcription factors, is known as a promoter [4]. Promoters are divided into a central core and several regulatory regions, usually in the 5′ region. The promoter core may also have a TATA box (consensus DNA sequence rich in adenine and thymine) and an initiator element that binds to transcription factors. It is already known that promoters can regulate the expression level of various transgenes and, in turn, these can be obtained from different sources; thus, their classification is divided into pol II (constitutive, inducible, and tissue-specific) and pol III (U3, U6); both are activated due to recognition by RNA polymerases [5].

On the other hand, the initiator elements can signal the start of the transcription in specific promoters that lack a TATA box [6]. The recognition of plant promoters would generally involve identifying and characterizing genes expressed in tissues or under conditions of some physiological stress [7] to identify promoters activated in these circumstances. The structurally characterized promoters could be fused with a coding sequence for later use in the genetic transformation of plants [8]; thus, the promoters are necessary to determine the expression of transgenes in genetically modified crops (Figure 1).

The study of promoters is necessary to understand the regulation of gene expression in plants [9]. The isolation of genes could be used as a starting point for identifying the promoters which drive the expression of the isolated gene. If the genome of the organism is already sequenced, basic local alignment tool (blast) analysis could be performed using the query of the sequence of the isolated gene; then, the location of the promoters will be at the 5′ region of the coding sequence of the gene. In the case that the genome of an organism is not sequenced, first, genes could be isolated through the creation of libraries, and once the gene sequence is known, the promoters of corresponding genes could be searched using genome walking strategies, such as PCR-based techniques [10,11]. The importance of isolating and identifying promoters lies in knowing which regions of interest are located on the 5′ side of a known genomic sequence. 

Whole-genome sequencing of many plants has facilitated the isolation of promoters from genomes, in which primers can be designed in the 5′ region of the open reading frame [12]. Once the promoter is isolated and sequenced, in silico analysis can be performed using bioinformatics tools such as PlantCARE [13], PlantProm DB [14], iProm-Zea [15], and TSSPlant [16]. Recent studies have shown that promoters play an essential role in processes such as gene editing with clustered regularly interspaced short palindromic repeats/CRISPR-associated protein 9 (CRISPR/Cas9), increasing the effectiveness and specificity in cutting double-stranded DNA. For example, in soybean, the function of the gRNA and Cas9 expressed under the control of the double 35S Cauliflower Mosaic Virus (CaMV) promoter was tested in hairy soybean roots where a high rate of mutations was obtained; which were even visualized in the next generation; opposite when using egg cell-specific promoters [17,18].

This review focuses on the identification of a wide variety of promoters, from constitutive, tissue specific, inducible, or synthetic, used in the expression of genes from different genetically modified plants due to the great boom they have for the development of agricultural and industrial products [1]. 

### 1.2. Types of Promoters

Promoters can be classified as constitutive, tissue-specific, inducible, and synthetic. The classification will depend on the function, type, and gene expression level [13,14].

### 1.3. Constitutive Promoters

Constitutive promoters are primarily used in plant transformation, promoting the expression of transgenes with different purposes, whether resistant to diseases, biotic stress, or insects [19]. One of the characteristics of the constitutive promoters is that they maintain control of the genes during most of the plant development; the expression levels will depend clearly on the type of cell with which they work. The 35S promoter derived from the CaMV is genetically modified crops’ most popular constitutive promoter (Table 1).

The function of the CaMV35S promoter was analyzed in cell colonies and in vitro banana plants by fusion with reporter genes (*luciferase* and *β-glucuronidase*) and the subsequent transformation of banana embryogenic cell suspensions (ECS) [38]. This promoter has been fused to the *luc2* reporter gene (isolated from the vector pGL4.10, Promega). Its activity has been characterized in cell lines and in vitro genetically modified banana plants [39,40]. The native banana promoter demonstrates high activity in ‘Williams’ banana plants in vitro (Figure 2). 

On the other hand, the characterization of the constitutive *actin* promoter demonstrated its efficacy in regulating genes in rice protoplasts when fused to the *β-glucuronidase* (*gus*) gene [20]. Like the study of ubiquitin genes, it has provided highly expressed plant constitutive promoters, especially in monocotyledonous plants isolated from rice [41] and maize [42].

### 1.4. Tissue-Specific Promoters

In the case of specific promoters, these direct the gene expression in a particular tissue, which can happen in different parts of the plant and at different stages [43]. In plant tissue, promoters can control/initiate the expression of a specific gene, such as in roots, seeds, or the vascular system [6].

For the expression of specific genes in genetically modified crops, it has been decided to use native or homologous promoters since they present a high specificity guaranteeing a correct genetic transformation [44]. In this concept, Manavella and Chan [45] mention the importance of using tissue-specific promoters since it would allow for more efficient crops [45] because the programmed regulation of the promoters is due to the correct expression of the transcription factors [43]. In this way, the study of vitamin A has been a public health problem worldwide, which is why Paul et al. [46] performed the characterization and isolation of two banana promoters of the *expansin1* gene (Exp1) and a fruit-specific *oxidase* (ACO) promoter for increasing vitamin A or retinol. These promoters were fused with the *gus* gene, and the activity of the fruit pulp was quantified by an ELISA assay, while fluorometric assays were used for the leaves and peel to determine the enzymatic activity. Finally, it was possible to determine that the prolonged maturation time of this fruit is essential to obtaining optimal concentrations of vitamin A. Few tissue-specific promoters have tangible expressions; for instance, Ye et al. [47] characterized two new cis-regulatory elements, GSE1 and GSE2. GSE1 activates promoters in all green plant tissues, while GSE2 is a regulator in the sheath and stem of plants, weakening gene expression [47]. 

Trivedi and Nath [48] performed the characterization and identification of the *Expansin* promoter for the first time, concluding that exposure to ethylene induces the overexpression of *MaEXPA1*, which progressively increases in the fruit and does not present expression in any other plant tissue. Furthermore, another four banana promoters were analyzed during maturation exposed to ethylene (*MaEXPA2*, *MaEXPA3*, *MaEXPA4*, and *MaEXPA5*) for the study in both dicots and monocots [48]. 

Only the *MaEXPA2* promoter was specific for the fruit and had a more robust expression, which is why it is recommended for application in monocots. In contrast, the other promoters were expressed in different plant tissues, although their competence could be analyzed in dicotyledonous plants [49]. The *MaEXPA1* promoter has already been characterized and is active in fruit tissue.

### 1.5. Inducible Promoters

Inducible promoters are activated by hormones, chemicals, environmental conditions, and biotic or abiotic stresses; and they contain a cis-acting element that could bind different transcription factors involved in the stimuli. In turn, the performance of inducible promoters is not usually affected by endogenous factors [50]. In the case of constitutive promoters, despite having a high and constant expression, they can trigger problems at the cellular level in non-specific places of the plant, which does not happen with an inducible promoter because a robust and temporal expression could be controlled depending on certain stimuli (Table 2) [43,51,52].

Investigation of promoters has identified those inducible to the attack of different pathogens. Barry et al. [70] evaluated the overexpression of inducible promoters closely linked to the *ACC oxidase* genes in tomatoes (*Solanum lycopersicum*). The AC1, AC2, and AC3 sequences were cloned and analyzed in various tissues and at different stages of plant development, demonstrating the high inducibility capacity in the development of tomato plants. AC1 and AC3 elements show accumulation during the senescence stage of leaves, fruits, and flowers. Santamaria et al. [71] have evaluated the resistance of the *A. thaliana* AtPRB1 promoter against different pathogens, for which the levels of ethylene and jasmonic acid increased in various stages of the plant in tissues such as root, stem, and flowers, being able to show high levels of overexpression of this promoter [71]. The CaIRL promoter was also identified from the isoflavone reductase gene from coffee (*Coffea arabica*). Expression studies showed that *CaIRL* fused to the *gus* reporter gene is exclusively expressed in coffee leaves, and the level of transcription increases markedly in response to biotic and abiotic stress, unlike that observed in healthy or unstressed plants [72]. Genes fused to inducible promoters could be expressed according to the promoter’s characteristics to specific stages of the development of an organism or a particular tissue under defined external conditions [52]. These promoters regulate the expression (mainly the activation, which can be switched from an OFF state to an ON state) of cloned genes in any organism by introducing the inducer. There are two ways in which the activity of a promoter can be regulated: positive and negative control (Figure 3).

### 1.6. Synthetic Promoters

Synthetic promoters consist of a core promoter and a combination of elements from diverse origins for spatial and temporal gene expression [73]. The effectiveness of these promoters lies in the expression patterns since they can be reformed to benefit either in type, some copies, or distance between motifs, the basis for the construction of synthetic promoters. On the other hand, they will differ significantly from native promoters because they can provide expression profiles that we cannot commonly find since they handle a combined profile of cis-elements, including enhancers, activators, or repressors from one core promoter sequence [74].

Due to the conditions of the native promoters, research has focused on designing new synthetic promoters with the capacity to perform gene transcription in a structured way and based on exposure to different stimuli [75]. The cis-acting elements in the promoter regions serve as biding sites for different transcription factors, which in turn are known to have the ability to modulate or regulate gene expression. Therefore, certain studies have also focused on specifically identifying each specific cis motif and determining the activity and regulation of each promoter with the aid of genetic engineering [76].

With this methodology, the activity of the CaMV35S promoter was increased by the fusion of cis-regulatory sequences from other promoters (CoYMV and CsVMV) [77]. Furthermore, because synthetic promoters are typically smaller in size and linked to regulatory element sequences, they must be carefully evaluated to reduce any interference or undesirable interaction between promoter sequences that are closely spaced [78]. In switchgrass green, three tissue-specific promoters have been characterized, for which an expression pattern was also evaluated. For the analysis of expression markers, the maize *ubiquitin 1* gene (ZmUbi1), switchgrass ubiquitin 2 (PvUbi2), and finally, the vector Cambia were used as positive controls [79].

On the other hand, an exhaustive analysis of genetically modified plants (cisgenic–intragenic) with native promoters is recommended since it would improve consumer acceptance of this product. The identification of native genes in cisgenic or intragenic cultures is carried out by expressing specific genes in the final product once the desired change has been obtained. Generally, the identified genes are linked to regulatory sequences, either promoters or terminators, being very beneficial to improve crops (Figure 4). 

Synthetic promoters possess cis-regulatory elements and have custom functionality ready to modulate their functions according to environmental stimuli [80]. Synthetic promoters combined with different transcription factors could provide the coordinated transcriptional control of multiple genes, which is necessary for successful metabolic engineering and implementing synthetic circuits in plants [73]. Therefore, other significant advantages of designing synthetic promoters for promoter activity could be modified either in a reduced or increased way only by modifying the cis-regulatory elements [75].

In this sense, bidirectional promoters could provide the expression of two genes simultaneously; increasing the number of genes that need to be expressed, Currently, there are several studies of bidirectional promoters in different species of model plants, such as *A. thaliana*, to improve the efficiency of CRISPR/Cas9 gene editing to be applied in commercial crops such as rice. The bidirectional (BiP) promoter was designed to express Cas9 and single guide RNA (sgRNA) in opposite directions, obtaining high levels of effectiveness compared to other systems [81]. With the aid of these promoters, it has been possible to investigate genes highly related to seed maturation conditions and ABA regulation [81]. There are other cases in which these promoters are expressed independently, as in the case of the fungus *Fusarium oxysporum*, where an intergenic region related to the regulation of hemicellulose degradation in plants was identified. This region was cloned with two reporter genes, namely *gus* and the enhanced green fluorescent protein (*egfp*), and tested independently [82].

### 1.7. Use of Reporter Genes in Plant Promoter Characterization 

Two main strategies could be used to study promoter activity. First, gene expression could be analyzed through the detection of mRNA with different techniques, including Northern blot, RT-(q)PCR, digital gene expression, microarray, and RNA-seq, among others. Therefore, the activity of the promoter of the corresponding gene is indicated once these techniques determine the gene expression pattern. The other strategy that is usually used to confirm promoter activity is the use of reporter genes. The use of reporter genes could be advantageous in the analysis of gene expression where the tissue could be challenging to obtain for the isolation of transcripts (e.g., trichomes, developmental seeds).

Reporter genes could be used to standardize plant genetic transformation protocols, the protein localization of plant tissues, and promoter characterization. In plants, the most used reporter gene is the *gus*. The use of the *gus* reporter gene was first reported by Jefferson et al. (1987) [83]. The *β-glucuronidase* gene (*gus*, *uidA*) was isolated from *E. coli*. Another reporter gene mostly used in plants is the *gfp* isolated from the Pacific jellyfish *Aequoria Victoria* [84,85]. Finally, the luciferase (*luc*) reporter gene from the American firefly *Photinus pyralis* is also used in plants [86].

To characterize promoters, researchers should choose a suitable reporter gene system according to the study’s objective. For instance, if spatial analysis of promoter activity is needed, the *gus* reporter gene is often used. Spatial analysis of promoter activity is usually detected when using the GUS reporter gene due to the long half-life (approximately 50 h) [83].

Therefore, careful analysis should be performed when screening at different developmental stages or under stress conditions due to the long half-life. For instance, the circadian promoter was not detected when using chloramphenicol acetyltransferase (*cat*) as a reporter gene, which has a half-life of 50 h, while a circadian pattern was detected for the same promoter fused to the luciferase reporter gene [87].

The GUS reporter system needs the use of a substrate (X-gluc), which is detrimental to in vivo analysis. On the other hand, the *gfp* avoids the use of exogenous substrates and is not invasive, although excitation is needed for fluorescence emission. Furthermore, tissue-specific expression, under specific stimuli, could be performed with the *gfp* reporter gene [85]. However, careful analysis of *gfp* expression is needed when assaying green tissues in plants, as background fluorescence is generated due to chlorophyll, which could be eliminated using proper filters [88,89,90].

The half-life of the modified eGFP is one day [91]; therefore, gene expression could be performed after one day when assaying temporal gene expression analysis. Furthermore, in vivo analysis could be performed using the *gfp* reporter gene. Another reporter gene in which in vivo analysis could be performed is the firefly (*Photinus pyralis*) *luc*, although a substrate should be added (luciferin).

Therefore, real-time gene expression analysis could be performed as LUC enzyme detection is non-invasive and non-destructive [86]. Inducible and developmental-regulated gene expression could be analyzed with LUC due to its short half-life (~15.3 min after luciferin is applied) [92]. However, light emission should be detected on specimens within a dark box and with a sophisticated camera system, charge-couple device (CCD) digital camera [93]. A luciferase reporter gene system on banana tissues is shown in Figure 5.

## 2. Conclusions

In recent years, transgenic techniques have improved several commercial crop species due to identifying many genes. Gene regulation and plant development depend mainly on the ability to control the expression of genes, which are highly related to promoters. As promoter studies advance, gene silencing, gene editing (e.g., CRISPR/Cas9), are processes that could express genes in a spatially and temporally specific manner, allowing more precise fits into the genome of cells [94]. This is why the field of research has focused above all on transgenesis methods because it is an advantageous technique for solving problems that are almost impossible with conventional breeding, hence the importance of identifying regulators or promoter elements [95].

Plant promoters contain different cis-elements, which contribute to regulating transcription. Sometimes, these cis-elements are expected to act individually or in groups to enhance or repress transcription. On the other hand, the exhaustive analysis of transcription factors allows us to obtain information on regulatory systems involved in responses to biotic or abiotic stress factors [73].

Gene expression must be strictly controlled due to the numerous interactions between genes. In this context, libraries of promoters with well-defined patterns of activity and functionality in plants should be characterized to continue with the study and analysis of different pathways for the genetic improvement of plants [96]. The design of new promoters to specifically express transgene expression is increasing as genetically modified crops need a more precise expression of improved traits.

## Figures and Tables

**Figure 1 genes-14-01226-f001:**
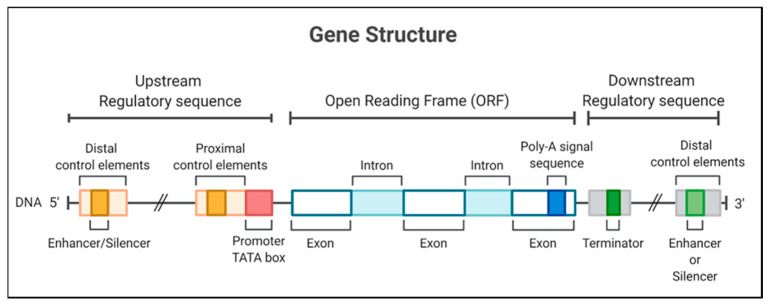
Gene structure, including the promoter region for a eukaryotic gene, encodes a protein. The gene diagrammed here contains a TATA box and different regulatory elements in 5′ and 3′ regions.

**Figure 2 genes-14-01226-f002:**
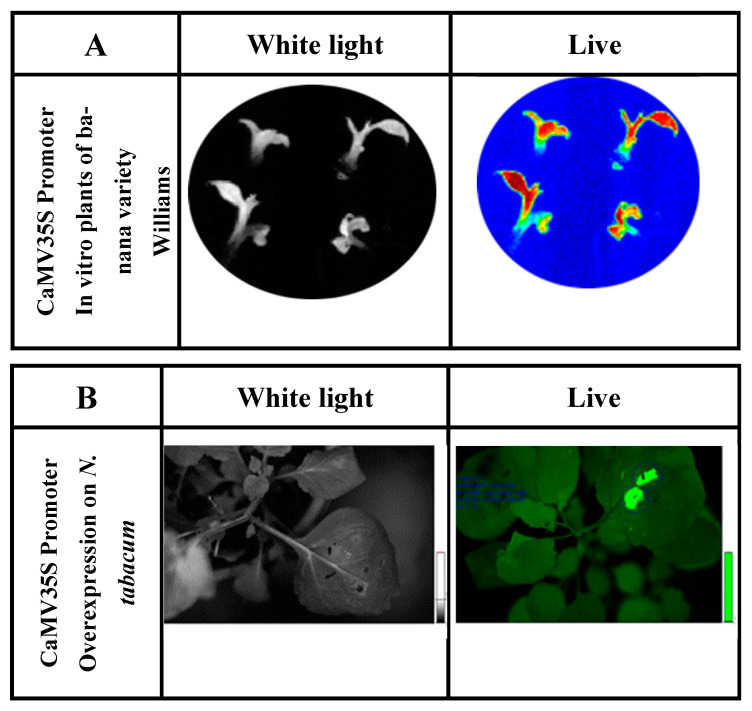
(**A**) Determination of luciferase reporter gene activity of in vitro ‘Williams’ banana plants transformed with the construct containing P35S::*luc2*. Exposure time was 3 min in total darkness; the photo was captured using the Stella 3200 equipment camera (Raytest, Straubenhardt, Germany). Luciferin dissolved in water (500 µm) was applied to in vitro plants. ‘Live’ indicates luciferase enzyme activity detected under conditions of total darkness. (**B**) Luciferase activity in *N. tabacum* leaves agroinfiltrated for the luciferase reporter gene expression fused to the CaMV 35S promoter (P35S::*luc2*).

**Figure 3 genes-14-01226-f003:**
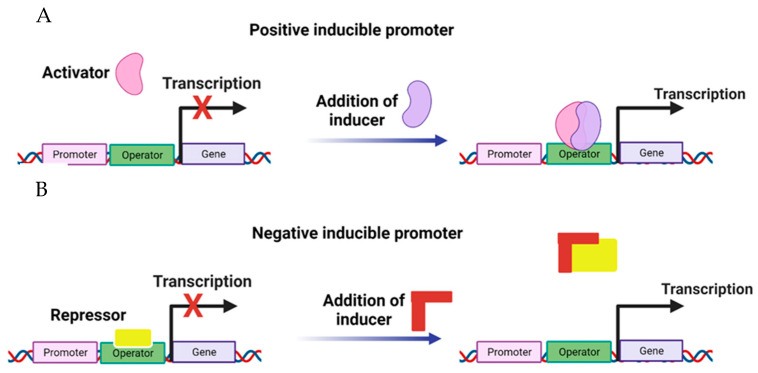
Inducible promoters. In positive induction, (**A**) promoter inactivity occurs due to an activator (OFF) absence. When stimulation occurs, the activator binds to the DNA, causing transcription (ON). In negative induction, (**B**) a repressor binds to DNA, blocking the transcription process (OFF). Then, by the action of an inducer, the repressor is released, initiating transcription (ON).

**Figure 4 genes-14-01226-f004:**
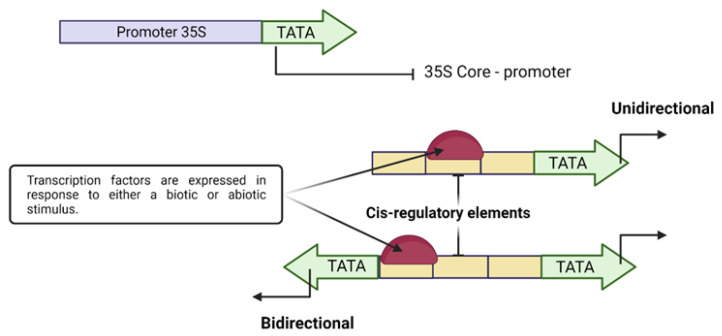
A synthetic plant promoter drives transcription, including the TATA box-containing core region of a constitutive wild-type promoter (CaMV 35S). A synthetic promoter comprises multiple copies of a cis motif upstream of the core 35S promoter for binding transcription factors expressed under different stimuli.

**Figure 5 genes-14-01226-f005:**
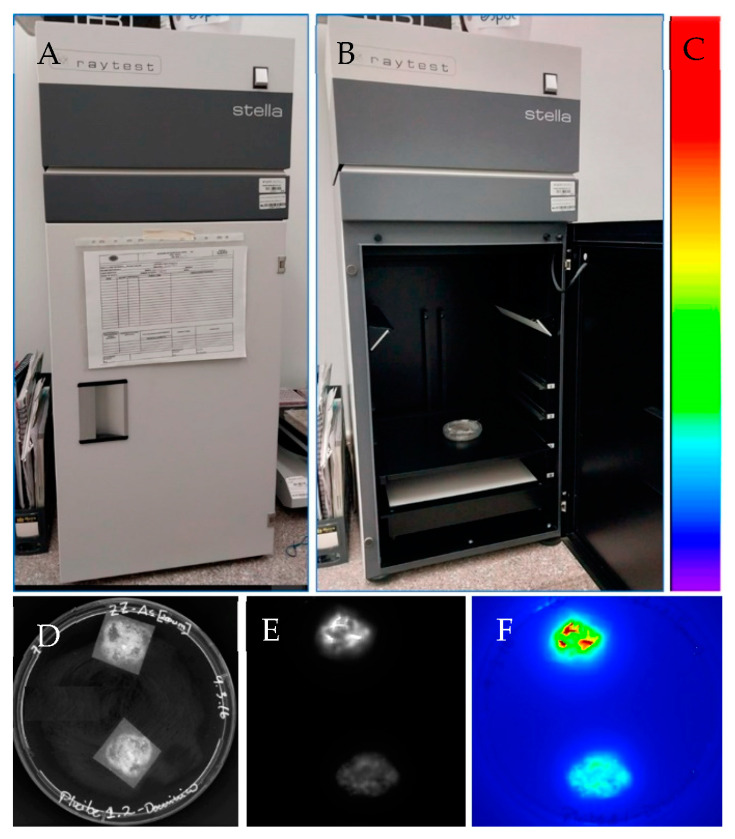
Luciferase reporter gene system for in vivo detection on banana tissues. (**A**) Light-tight box linked to a CCD camera used for luciferase activity detection in vivo (Stella 3200, Raytest, Germany). (**B**) Sample (Petri dish) containing banana embryogenic cells within the light-tight box. (**C**) Colors indicate luciferase activity. Red indicates high luciferase activity while blue indicates no luciferase activity. (**D**) Banana embryogenic cells after 3 months of *Agrobacterium*-mediated transformation using the strain EHA105 harboring the pLVCIBE2 vector (P35S::*luc2*). Picture was captured in the light-tight box (Stella 3200) under light conditions. The construction of the pLVCIBE2 was as follows: the *luc2* was obtained from the pGL4 plasmid (Promega). The *gus* reporter gene (*uidA^INT^*) was digested with the enzymes NcoI (C^CATG_G) and BstEII (G^GTNAC_C) from the pCAMBIA 1301. PCR was performed using the Expand High Fidelity mix (Roche) for the amplification of the *luc2* from pGLA4. The primers used were designed with the respective restriction enzyme recognition sequence at the 5′ end (Forward: TAGTACCATGGGGTAAAGCCACCATGGAAGA; reverse: TAGTAGGTCACCCCGCCCCGACTCTAGAATTA). *Agrobacterium*-mediated transformation was performed according to Santos et al. [38] with some modifications. Briefly, ECS from the banana cultivar ‘Williams’ were developed from male inflorescences. A 33% settled cell volume of 200 μL of ECS (which correspond to approximately 50 mg of fresh banana cells) were co-cultured with 1000 μL of acetosyringone-induced *Agrobacterium* at OD_600nm_ of 0.4 for 6 h in darkness in a shaker at 25 rpm. Then, banana cells were collected using a 200 μM polyester mesh and subcultured in a ZZ medium for 7 days. Later, cells were subcultured on ZZ medium supplemented with 12.5 mg/L hygromycin and 200 mg/L Timentin®. After one month in a selection medium, luciferase activity was measured. (**E**) Luciferase activity was detected using the Stella 3200 after adding 20 μL luciferin (500 μM) to the sample, with an acquisition time of 1 min in complete darkness. Image is in greyscale. (**F**) Acquired image was transformed to pseudo colors.

**Table 1 genes-14-01226-t001:** Constitutive promoters are used in the generation of genetically modified plants.

Promoter	Source	Expression	References
*Act1*	Actin gene, rice	The whole plant, preferably monocots	[20,21]
*Adh1*	Alcohol dehydrogenase gene, maize	Roots, meristematic tissue, endosperm, and pollen (anaerobic regulation) preference in monocots	[22,23]
*HSP18.2*	*Arabidopsis thaliana*	Leaves, vascular system	[24]
*ScBV*	Bacilliform virus, sugarcane	Leaves, vascular system, monocots, and dicots	[25,26]
*Ubi-1*	Ubiquitin gene, maize	Protoplast, monocots.	[27]
*RUBQ1/RUBQ2*	Ubiquitin gene, rice	Genes expression, monocots.	[28]
*Gmubi*	Soybean	The whole plant	[29]
CaMV35S	Cauliflower mosaic virus	Expression throughout the plant, monocots, and dicots	[30]
*nos*	Nopaline synthase gene, *Agrobacterium tumefaciens*	Expression throughout the plant, monocots, and dicots	[31,32]
CmYLCV	Cestrum Yellow Leaf Curling Virus (CmYLCV)	Growth and development	[33]
*KST1*	*Solanum tuberosum* (potato)	Guard cell promoter	[34]
*Cula11/Cula08*	*Cunninghamia lanceolate* (Chinese fir)	Protoplast, monocots, and dicots	[35]
*P OsCon1*	Rice	The whole plant, monocots, and dicots	[36]
*TCTP*	Oil palm	Immature embryo, embryogenic callus, embryoid, a young leaflet from a mature palm, green leaf, mesocarp, and stem	[37]

**Table 2 genes-14-01226-t002:** Tissue-specific and inducible promoters in plants.

Type of Promoter	Promoter	Source	Expression	References
Tissue specific	*β-phaseolin*	*Phaseolus vulgaris* (phas)	Flowers, seeds, embryogenesis	[53,54]
	*EXP1*	Banana	Ripening fruit	[46,48,49,55]
	*GSSP1, GSSP3, GSSP5, GSSP6, GSSP7*	*Oryza sativa*	Bidirectional green tissue	[47]
	*MT3-A*	Oil palm	Mesocarp	[56]
	*LC01*	Oil palm	Leaf specific	[57]
	*SynR2 SynR1*	*N. tabacum*	Root	[58]
Inducible	*pCL*	*S. tuberosum*	Gene regulation of the activity of acid vacuolar invertase in potato tubers at low temperature	[59]
	*LA22CD07, LesAffx.6852.1.S1_at*	Tomato ripening-induced genes	Chemical factors induce fruit ripening	[60]
	*POD, POX*	Oxidative stress-inducible peroxidase	Rice peroxidase inhibitor, biotic stress	[61,62]
	*PR-1a*	*A. thaliana*	Related to pathogenesis, the acquired resistance system (SAR)	[63]
	*GST1*	*Potato*	Biotic stress	[64]
	*SGD24-STR246C*	*Tobacco*	Biotic stress	[65]
	*Zmap*	*Maize*	Different stressors	[66]
	* CMPG1 *	*A. thaliana*	Pathogen inducible	[67]
	*Synp16*	*Soybean*	Abiotic stress	[68]
	*GCC*	*A. thaliana*	Jasmonic acid inducible	[69]

## Data Availability

All data is available in the manuscript.

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
