# Peer review of "Plant Promoters: Their Identification, Characterization, and Role in Gene Regulation"

_genes, 2023, doi:10.3390/genes14061226_

Round 1

Reviewer 1 Report

Ms. ID genes-2315850

Title: The role of promoters in the identification, characterization, and gene regulation in plants

Submitted to section: Plant Genetics and Genomics

This review is devoted to the important problem of the diversity of plant gene promoters and their use to improve the function of genes in plants. The authors attempted to describe the types of promoters: constitutive, tissue-specific, inducible, and synthetic. They cited 61 publications on the topic, a part of them of recent years.

However, on the basis of the criteria provided to me in considering the current paper for publication, I feel that this paper is not acceptable for publication in an international level journal and, particularly, in “Genes”, as it stands. First, it is not clear what new this review has brought to the vision of the chosen topic in comparison with the available reviews. For example, the review on this topic Porto et al., 2014, cited in this article (see reference 5), describes the main positions much better. The role of promoters in the identification, characterization, and regulation of genes in plants, stated in the title of the article, is described rather vaguely. The authors gave a classification of promoters and several examples for each type. The article lacks the rigorous analysis of the problem required for publication.

In addition, the assignment of the described promoters to a particular type sometimes raises questions. For example, in Table 1, HSP1&2 is classified as “constitutive”, but “expression” is described as “heat-shock and stress in plants”; pKST1 is classified as “constitutive” (not “specific”), but “expression” is described as “first guard cell promoter”; ADH1 is classified as “constitutive”, but expression of the gene under normal conditions is localized in certain cell types and strongly induced under oxygen deprivation etc.

Given the topic of the article, ‘promoter’ should be added to the list of keywords.

The quality of the text is not high enough. For example:

Line 24: “…constitutive tissue, cell-specific … promoters” – Are the authors sure the punctuation marks are correct, if they distinguish between constitutive and tissue-specific types?

Line 60: The sentence about PCR is not clearly linked to the following text.

Line 71-74: It is better to replace this part to the Introduction.

Line 71: “Sustainable” is repeat twice. The sentence needs correction.

Line 102-103: This information is present in the title of Figure 2 and is not needed in the text. Obviously, such detailed information is redundant for a review. Add a link for this figure.

Line 107: The phrase “promoters can express… a gene” is not the best. Promoters control / initiate the expression of a gene.

Line 158-159: The sentence is unclear.

Line 139-140 and 140-141: These two sentences overlap (“biotic and abiotic factors”). And hormones are endogenous factors (compare lines 139 and 141).

Line 192-194: Two similar uninformative sentences that are weakly related to the figure 4.

Author Response

Dear Editor

We appreciate all the comments from the reviewers. All the comments were addressed and are indicated below:

Reviewer 1:

Line 24: “…constitutive tissue, cell-specific … promoters” – Are the authors sure the punctuation marks are correct, if they distinguish between constitutive and tissue-specific types?

Response: Sentence was eliminated

Line 60: The sentence about PCR is not clearly linked to the following text.

Response: The following sentence was added:

The isolation of genes could be used as a starting point for the identification of the promoters which drives the expression of the isolated gene. In the case that the genome of the organism is already sequenced, blast analysis could be performed using as query the sequence of the isolated gene; then, the location of the promoters will be at the 5´region of the coding sequence of the gene. In the case that the genome of an organism is not sequenced; first, genes could be isolated through the creation of libraries, and once the gene sequence is known, the promoters of corresponding genes could be searched using genome walking strategies, such as PCR-based techniques [7, 8].

Line 71-74: It is better to replace this part to the Introduction.

Response: Sentence was replaced

Line 71: “Sustainable” is repeat twice. The sentence needs correction.

Response: Sentence was corrected

Line 102-103: This information is present in the title of Figure 2and is not needed in the text. Obviously, such detailed information is redundant for a review. Add a link for this figure.

Response: Sentence was eliminated.

Line 107: The phrase “promoters can express… a gene” is not the best. Promoters control / initiate the expression of a gene.

Response: Sentence was modified

Line 158-159: The sentence is unclear.

Response: Sentence was modified

Line 139-140 and 140-141: These two sentences overlap (“bioticand abiotic factors”). And hormones are endogenous factors(compare lines 139 and 141).

Response: Sentences were modified

Line 192-194: Two similar uninformative sentences that are weakly related to the figure 4.

Response: Sentences were modified

Reviewer 2 Report

This study provides The role of promoters in the identification, characterization, and gene regulation in plants which would be helpful for future studies and management of crop breeding. However, there are some shortcomings which must be revised.

The abstract should contain brief methods and main findings.

Also add significance of the study in the abstract.

Introduction of the topic is missing it should be included by discussing the background, significance, impacts of this study, and aim of the study.

Line 54-59 provide a complete overview of the studies and techniques one is not enough.

All genes names must be italicize.

Include future recommendations in the conclusion.

Author Response

Dear Editor

We appreciate all the comments from the reviewers. All the comments were addressed and are indicated below:

REVIEWER 2

This study provides “The role of promoters in the identification, characterization, and gene regulation in plants which would behelpful for future studies and management of crop breeding. However, there are some shortcomings which must be revised.

The abstract should contain brief methods and main findings.

Response: Abstract was modified

Also add significance of the study in the abstract.

Response: Abstract was modified

Introduction of the topic is missing it should be included by discussing the background, significance, impacts of this study, and aim of the study.

Response: First paragraph was added and fourth paragraph was modified

Line 54-59 provide a complete overview of the studies and techniques one is not enough.

Response: Paragraph was modified.

All genes names must be italicize.

Response: gene names were italized

Include future recommendations in the conclusion.

Response:  Recommendations was added

Reviewer 3 Report

The manuscript stresses the importance of promoters in plant biotechnology. Improved crop varieties are used to overcome diseases and stress. Genetic improvement can be done through breeding, mutation, transformation, or gene editing. Promoters are important for regulating gene expression and improving traits in transgenic crops. Promoter analysis helps generate biotech crops via genetic transformation. Different promoter types (constitutive, cell-specific, inducible, synthetic) have been discovered, aiding gene regulation and plant development. Promoters are vital for ensuring correct gene expression in biotech processes.

Suggestions:

Moderate English language corrections are suggested to the authors.

Author Response

Dear Editor

We appreciate all the comments from the reviewers. All the comments were addressed and are indicated below:

Reviewer 3

The manuscript stresses the importance of promoters in plantbiotechnology. Improved crop varieties are used to overcome diseases and stress. Genetic improvement can be done through breeding, mutation, transformation, or gene editing. Promoters are important for regulating gene expression and improving traits in transgenic crops. Promoter analysis helps generate biotech crops via genetic transformation. Different promoter types (constitutive, cell-specific, inducible, synthetic) have been discovered, aiding gene regulation and plant development. Promoters are vital for ensuring correct gene expression in biotech processes.

Suggestions:

Moderate English language corrections are suggested to the authors.

Response: English was improved

Reviewer 4 Report

The manuscript of Villao et al. reviews the role of promoters in gene regulation in plants. The whole manuscript has been written well except few exceptions. That being said the review does not include any new information that was not compiled and published before by other authors.

Some of the comments or suggestions are given below-

-       The title of the review is incomplete and confusing. It reads “The role of promoters in the identification, characterization, and gene regulation in plants”, so the role of promoters in the identification and characterization of what? I would suggest to rephrase or modify the title to make it more meaningful.

-       The review did not include all the promoters playing role in gene regulation in plants. I would suggest to make it more comprehensive covering all the promoters regulating the gene expression in plants.

-       The authors mentioned only few examples of promoters, I would suggest authors to think about putting separate tables for each type of promoters and put more examples for each type.

-       As I said earlier, the review is well-written except few sentences, for example, line 71-72 need to be rephrased to make it more understandable.

-       I don’t find much new/extra information that can make the review different from the review published by Kummari et al (2020).

I would suggest accepting the paper with major revision. I strongly encourage the authors to resubmit a REVISED manuscript including all the latest information regarding plant promoters regulating the gene expression to make it a comprehensive review and worth publishing.

Author Response

Dear Editor

We appreciate all the comments from the reviewers. All the comments were addressed and are indicated below:

Reviewer 4

The manuscript of Villao et al. reviews the role of promoters in gene regulation in plants. The whole manuscript has been written well except few exceptions. That being said the review does not include any new information that was not compiled and published before by other authors.

Some of the comments or suggestions are given below-

- The title of the review is incomplete and confusing. It reads “The role of promoters in the identification, characterization, and gene regulation in plants”, so the role of promoters in the identification and characterization of what? I would suggest to rephrase or modify the title to make it more meaningful.

Response: Title was modified

- The review did not include all the promoters playing role in gene regulation in plants. I would suggest to make it more comprehensive covering all the promoters regulating the gene expression in plants.

Response: More promoters were added (Tables 1 and 2)

-The authors mentioned only few examples of promoters, I would suggest authors to think about putting separate tables for each type of promoters and put more examples for each type.

Response: More promoters were added (Tables 1 and 2)

- As I said earlier, the review is well-written except few sentences, for example, line 71-72 need to be rephrased to make it more understandable.

Response: Sentence was added to the introduction part

- I don’t find much new/extra information that can make the review different from the review published by Kummari et al. (2020).

Response: The topics covered from Kummari et al. (2020) are not the same with the manuscript submitted. It covers new data, for instance the banana promoter and use of the luciferase reporter gene for it characterization (Villao et al. 2019, 2021).

Round 2

Reviewer 1 Report

Ms. ID genes-2315850

Title: The role of promoters in the identification, characterization, and gene regulation in plants

Revised version

Submitted to section: Plant Genetics and Genomics

The authors took into account all my comments, and, in my opinion, this version of the manuscript is suitable for publication in Genes.

Reviewer 4 Report

The authors addressed many issues/concerns I raised, and the revised version has been improved and has now better flow of reading. I would suggest to accept the review paper for publication in the Genes journal with minor revision.

Having said that there are still some corrections/suggestions highlighted in the manuscript that need to be corrected (see the attached manuscript).

-       The title of the manuscript is still incomplete, the title can be modified as “Plant Promoters: Identification, Characterization and their Role in Gene Regulation”. It’s just a suggestion, I would leave it to authors and editor to finalize the title.

-       The figure 2 needs to be corrected. You will still the show/hide symbols in the figure.  

-       The author put the tissue-specific and inducible plant promoters in the same table (#2). I would suggest to put sub-headings to separate these two types of promoters.

-       I also want to mention that it is very difficult to read and track the sentences in the manuscript with so many corrections in the text. I believe the Editor/English editor will take care of that.
